# Long-Term In Vitro Adhesive Properties of Two Universal Adhesives to Dentin

**DOI:** 10.3390/ma16093458

**Published:** 2023-04-28

**Authors:** Ana Hurtado, Victoria Fuentes, María Cura, Aitana Tamayo, Laura Ceballos

**Affiliations:** 1International Doctoral School, Rey Juan Carlos University, 28008 Madrid, Spain; ana.hurtadofernandez@gmail.com; 2IDIBO Research Group, Faculty of Health Sciences, Rey Juan Carlos University, 28922 Alcorcón, Madrid, Spain; laura.ceballos@urjc.es; 3Faculty of Health Sciences, Rey Juan Carlos University, 28922 Alcorcón, Madrid, Spain; odontologia@mariacura.com; 4Institute of Ceramics and Glass, CSIC, Kelsen 5, 28049 Madrid, Spain; aitanath@icv.csic.es

**Keywords:** universal adhesives, adhesion, bond strength, nanoleakage, degree of conversion, dentin

## Abstract

The composition of universal adhesives, as well as the adhesive strategy, may influence bonding effectiveness and durability. This study aimed to evaluate the microtensile bond strength (µTBS) and nanoleakage (NL), immediately and after 6-month aging, and in situ degree of conversion (DC), of two universal adhesives (Scotchbond Universal Adhesive, SBU; Xeno Select, XEN) applied with etch-and-rinse (ER) and self-etch (SE) strategies, in comparison with a two-step SE adhesive (Clearfil SE Bond, CSE). Dentin surfaces of fifty human third molars were randomly assigned to the following adhesives: two universal adhesives, SBU and XEN, applied in ER or SE mode and CSE, used as control. Teeth were sectioned into beams to be tested under µTBS, half of them after 24 h, and the rest after 6 months of water aging. Selected beams from each tooth were used for NL evaluation and in situ DC quantification. SBU and CSE showed significantly higher mean µTBS and lower nanoleakage than XEN, regardless of the evaluation time and adhesion strategy. XEN-SE yielded the lowest degree of conversion. Therefore, adhesive properties of universal adhesives to dentin are material dependent, regardless of the adhesion strategy, exhibiting XEN a significantly worse performance than SBU.

## 1. Introduction

Dentin adhesives have continuously been developed to simplify adhesive protocols and to reduce technique sensitivity being the universal adhesives or multimode adhesives the newest generation [1]. Universal adhesives are basically SE systems, in terms of adhesive composition and bonding procedures [2], with the advantage that they can be used in SE or ER, or with selective enamel ER mode, depending on the clinician’s preference [3,4].

The optional adhesion strategy in universal adhesives to dentin is possible through the inclusion of functional monomers that play a major role in chemical adhesion to this substrate. The most common functional monomer added is 10-methacryloyloxydecyl dihydrogen phosphate (10-MDP). This monomer has a proven potential to chemically interact with hydroxyapatite creating a nano-layered structure of MDP-Ca salts at the interface that appears to be hydrolytically stable and improves adhesion strength [1,3,5]. Alternatively, other functional monomers can be found in the composition of these universal adhesives, such as dipentaerythritol pentaacrylate monophosphate (PENTA) or glycero-phosphate dimethacrylate (GPDM), although their key contribution to the adhesive properties to dentin remains controversial [2,6,7].

Moreover, universal adhesives were simplified by incorporating hydrophilic monomers, mainly HEMA (2-hydroxyethyl methacrylate), water, and other solvents, to improve the bond to moist dentin. Nevertheless, the stability of dentin-adhesive interfaces generated by these adhesive systems continues to be questionable due to the presence of both hydrophobic and hydrophilic components on the same bottle [8]. HEMA is frequently added as an essential component to prevent this phase separation between other monomers and water [9,10]. Moreover, its low molecular weight and high hydrophilicity promote surface wetting and penetration into the demineralized dentin [11,12]. However, several drawbacks have been associated with HEMA, hereby affecting bond durability. HEMA does not effectively contribute to the polymerization of the resin matrix and, thus, to the mechanical properties of the formed polymer matrix [10]. Its hydrophilicity makes it susceptible to hydrolysis, water sorption, and hydrolytic degradation of the adhesive layers [13]. Furthermore, HEMA inhibits the formation of 10-MDP-Ca-salt nanolayering at SE adhesive interfaces [14], and it is also known to give rise to allergic reactions [3].

Scotchbond Universal Adhesive (3M Oral Care) was the first universal adhesive system launched into the market and the most prevalent universal adhesive in the peer-reviewed literature [2,4,8,15,16]. It contains 10-MDP, ethanol, and HEMA, and its bonding effectiveness has been tested in vitro and in clinical studies, exhibiting good results. Other universal adhesives have already been marketed which are totally HEMA-free and include other methacrylic acid-based monomers. Although one of the first adhesives to be available in the market was Xeno Select (Dentsply Sirona), limited information is available on the durability of bonding effectiveness of this 10-MDP-free and HEMA-free universal adhesive [17,18,19,20,21,22,23,24]. In vitro, its bonding ability to dentin seems not to depend on the adhesion strategy applied [21]. Therefore, the aim of this study was to determine the microtensile bond strength (µTBS) and nanoleakage (NL), immediately and after 6-month aging, as well as in situ degree of conversion (DC) of two universal adhesives with different formulations, (Scotchbond Universal Adhesive, that contains 10-MDP and HEMA, and a 10-MDP and HEMA-free adhesive, such as Xeno Select), using ER and SE strategies, in comparison to a two-step SE adhesive (Clearfil SE Bond, Kuraray).

The null hypotheses tested were as follows: (1) Microtensile bond strength to dentin of two universal adhesive systems used with ER and SE strategies is similar to that obtained with a two-step SE system, immediately and after 6-month water aging. (2) The sealing ability of these universal adhesives, applied in ER and SE modes, is not different from the one registered for a two-step SE system, at both evaluation times. Moreover, (3) The in situ degree of cure of the adhesives mentioned is comparable, regardless of the adhesion strategy used.

## 2. Materials and Methods

Fifty extracted non-carious human third molars, stored in 0.1% thymol solution at 4 °C, were used within six months after extraction. Teeth were collected after protocol approval by the ethics committee of Rey Juan Carlos University (Madrid, Spain). 

The occlusal enamel was perpendicularly sectioned to the long axis of the tooth using a slow-speed diamond saw (Isomet 5000, Buehler; Lake Bluff, IL, USA) to expose a superficial dentin surface surrounded by enamel. This surface was then ground using 600-grit wet SiC abrasive papers for 60 s to standardize the smear layer. Moreover, the absence of enamel was verified under a light stereomicroscope (Olympus SZX7, Hamburg, Germany).

### 2.1. Experimental Groups and Design

Teeth were randomly assigned to four experimental groups (n = 10 teeth) according to the universal adhesive system and the adhesion strategy applied (ER or SE). The adhesives tested were Scotchbond Universal Adhesive (SBU; 3M Oral Care, St. Paul, MN, USA, also known as Single Bond Universal in some countries) and Xeno Select (XEN; Dentsply Sirona, Konstanz, Germany). Therefore, both were evaluated as two-step ER adhesives (SBU-ER and XEN-ER) and as one-step SE adhesives (SBU-SE and XEN-SE). The two-step SE adhesive, Clearfil SE Bond (CSE; Kuraray, Noritake Dental Inc., Tokyo, Japan) was used as the control group. Adhesive systems were all applied on dentin surfaces following respective manufacturers’ instructions and light-cured for 10 s using a LED unit set at 800 mW/cm^2^ (Bluephase, Ivoclar Vivadent, Schaan, Liechtenstein). Composite buildups were created by applying three incremental layers (2 mm each) of a light-cured universal hybrid resin composite (Filtek Z250, 3M Oral) (Table 1).

After storage in distilled water for 24 h at 37 °C, the assemblies were longitudinally sectioned in “x” and “y” directions across the bonded interface with a slow-speed diamond saw (Isomet 5000) to obtain resin–dentin beams with a cross-sectional area of approximately 0.5 ± 0.2 mm^2^ measured with a digital caliper (Digimatic Caliper, Mitutoyo, Tokyo, Japan).

Half of the specimens were randomly selected from each tooth to be stored in distilled water for 24 h at 37 °C, and the other half were kept for 6 months in distilled water at 37 °C. After each storage period, specimens were submitted to microtensile bond strength (µTBS) and nanoleakage (NL) tests. Two resin–dentin bonded beams from each tooth were previously selected for the degree of conversion in situ measurements (DC). The specimen preparation protocol is schematically presented in Figure 1.

### 2.2. Microtensile Bond Strength (µTBS)

For the µTBS testing, the ends of the specimens were carefully fixed with cyanoacrylate glue (Loctite Super Glue-3 gel, Henkel, Düsseldorf, Germany) to a jig in a universal testing machine (Instron 3345, Instron Co., Canton, MA, USA) and stressed at a crosshead speed of 0.5 mm/min until failure. The µTBS values were calculated in MPa by dividing the load at failure by the cross-sectional bonding area. Fractured beams were observed by a single operator using a stereomicroscope at 40× magnification to determine the mode of failure: adhesive (A, between resin composite and dentin), cohesive (CD, in dentin; CC, in resin composite), or mixed (M, simultaneous adhesive and cohesive failures). 

### 2.3. Nanoleakage (NL)

Two beams, obtained from each tooth at both storage times, were coated with two layers of nail varnish applied 1 mm from the bonded interfaces. After rehydration in distilled water for 10 min, specimens were placed in a 50 wt% ammoniacal silver nitrate solution (pH = 9.9) in darkness for 24 h at 37 °C. Specimens were rinsed thoroughly in distilled water for 1 min and immersed in a photo-developing solution for 8 h under fluorescent light to reduce silver ions into metallic silver grains within voids along the bonded interface. 

Then, specimens were wet-polished with a 600 SiC paper to remove the nail varnish, and then, embedded in epoxy resin (Epo-Thin 2, Buehler). After setting, the bonding interfaces were exposed and polished with 1000-grit SiC paper and 6, 3, 1, and 0.25 μm diamond pastes using a polishing cloth. All specimens were then ultrasonically cleaned in distilled water for 10 min, air-dried, and mounted on aluminum stubs. 

Resin–dentin interfaces were analyzed in a field-emission scanning electron microscope (Philips XL30 SEM, FEI Company, Hillsboro, OR, USA) operated in the backscattered electron mode using energy dispersive X-ray spectrometry (EDX). The amount of silver nitrate within the adhesive layer, hybrid layer, and resin tags in each beam was measured with EDX in three regions (5 × 5 μm) of the bonded interfaces (left, center, and right). The percentage distribution of metallic silver particles at the adhesive/tooth interface was calculated with a digital image-analysis software (Photoshop CC software version 19, Adobe Inc., Mountain View, CA, USA) in a selected area on each image at 5000×. Additional micrographs were made at 2500× to characterize the NL pattern.

### 2.4. Degree of Conversion In Situ (DC)

Two resin–dentin bonded beams from each tooth were wet polished with 1500, 2000, and 2500 grit SiC papers for 15 s. Then, they were ultrasonically cleaned for 20 min and stored in distilled water for 24 h at 37 °C. The micro-Raman spectrometer (Renishaw InVia, Renishaw plc, Gloucestershire, UK) was first calibrated for zero and then for coefficient values using a silicon specimen (line at 520 cm^−1^). The following micro-Raman parameters were employed: 20 mW neon laser with 532 nm wavelength, a spatial resolution of 3 mm, spectral resolution 5 cm^−1^ accumulation time of 30 s with 6 co-additions, and magnification of 100× (Leica DM microscope, Leica Microsystems Wetzlar GmbH, Wetzlar, Germany) to a 1 mm beam diameter. Spectra were taken at the dentin-adhesive interface at three different sites for each specimen and an average per tooth was calculated. Spectra of uncured adhesives were taken as reference. Post-processing of spectra was performed using the dedicated Opus Spectroscopy Software version 6.5 (Bruker Optik GmbH, Ettlingen, Baden Württemberg, Germany). The ratio of double-bond content of monomer to polymer in the adhesive was calculated according to the following formula: DC (%) = (1 − R_(cured)_/R_(uncured)_) × 100
where R is the ratio of aliphatic and aromatic peak areas at 1639 cm^1^ and 1609 cm^1^ in cured and uncured adhesives [25].

### 2.5. Statistical Analysis

For each adhesive, the µTBS values of all beams from the same tooth were averaged for statistical purposes. Thus, the tooth was considered the experimental unit and 10 teeth was the size of each experimental group, according to the adhesive system at each evaluation time (24 h and 6-month water storage). PTFs were included in the statistical analysis as 0 MPa and equally distributed to both evaluation times. Specimens with cohesive fractures were discarded for the µTBS. The influence of the adhesive system and evaluation time on µTBS was analyzed using a two-way ANOVA and Tukey’s multiple comparisons tests. Previously, the normal distribution of this variable was confirmed as well as homoscedasticity by Shapiro–Wilk and Levene tests, respectively (*p* > 0.05). For each adhesive, the comparison of the mean µTBS at 24 h and 6 months was carried out with Student’s *t*-test for paired data. NL data were not normally distributed (Shapiro–Wilk test, *p* < 0.05) and Kruskal–Wallis and the Mann–Whitney U tests were performed, followed by Bonferroni correction at both evaluation times. Comparisons of NL data between 24 h and 6 months for each experimental group, were performed using Wilcoxon test. Immediate results of DC in situ were analyzed by one-way ANOVA and Tukey’s multiple comparisons tests. All statistical tests were performed at a pre-set alpha of 0.05 by means of IBM SPSS 20 (IBM Corporation, Armonk, NY, USA).

## 3. Results

### 3.1. Microtensile Bond Strength (µTBS)

Descriptive data of µTBS for each experimental groups are shown in Table 2. 

Both universal adhesives showed pre-test failures although the percentage was higher when XEN was used. Two-way ANOVA revealed that µTBS values were significantly influenced by the adhesive system (*p* < 0.001). However, evaluation time (*p* > 0.05) and the interaction between the adhesive system and aging time (*p* > 0.05) did not significantly affect dentin bond strength. For both evaluation times, significantly higher mean µTBS values were achieved with SBU using both strategies but similar to those of CSE. Both, XEN in ER and SE modes, showed statistically lower mean μTBS results than those of SBU and CSE without significant differences between them.

After 6 months of water storage, bond strength values decreased for specimens bonded with XEN. In contrast, the results remained stable for the SBU-SE and CSE groups, and even improved in the SBU-ER group.

The mode of failure per group is also displayed in Table 2 and failures were predominantly adhesive for all groups. Mixed failures were not detected for any of the adhesive systems tested. 

### 3.2. Nanoleakage (NL)

Regardless of the adhesive system and strategy used, all interfaces showed nanoleakage (Table 3). 

However, it was significantly influenced by the experimental group at each aging time (24 h, *p* = 0.004; 6 months, *p* < 0.001). After 24 h, SBU-ER and SBU-SE achieved similar NL values to those of CSE. Both XEN groups exhibited higher NL values than CSE, and similar to those of SBU. Thus, the adhesion strategy did not influence the values obtained. 

At 6 months, both XEN groups yielded higher NL values than SBU-SE and CSE. In contrast, aging time had a significant influence on NL for SBU-SE (*p* = 0.028), as a better sealing ability was detected after 6 months of storage. In the other groups, the percentage of nanoleakage remained statistically stable.

Representative micrographs of the bonding interfaces for all groups are depicted in Figure 2 and Figure 3. 

Low levels of silver deposits were detected for SBU and CSB groups. For the former, fibrillary extensions were observed above the hybrid layer regardless of the adhesion strategy. Both XEN groups exhibited greater nanoleakage with a characteristic water tree pattern of HEMA-free adhesives.

### 3.3. Degree of Conversion

Mean values and standard deviations of degree of conversion values obtained for each experimental group are shown in Table 3. One-way ANOVA revealed that the adhesive system significantly influenced in situ DC results (*p* < 0.001). Similar values were determined for both strategies of SBU and for XEN-ER and CSE. However, XEN attained statistically lower DC when applied in the SE strategy.

## 4. Discussion

The present study examined the adhesive properties of two universal adhesives applied in ER and SE strategies at different evaluation times. According to the results, the null hypotheses were rejected, since XEN showed the worst microtensile and nanoleakage values regardless of the adhesion strategy applied and aging period, as well as the lowest DC percentage in SE mode. However, comparable results were obtained by SBU and CSE for all tests and experimental conditions. 

In the present study, the two-step self-etch adhesive CSE was used as control, since it has been considered the gold standard for dentin bonding [3]. This adhesive contains 10-MDP monomer, patented by Kuraray, which was added to the composition of most universal adhesives, such as SBU, once the patent expired. Moreover, CSE is not a simplified adhesive, as requires the application of a separate hydrophobic bonding layer, which reduces the fluid flow across the interface [26], and the concentration of unreacted monomers and retained solvents in the adhesive layer. This contributes to improve the degree of conversion and hydrolytic degradation resistance, and longevity [27,28,29], in accordance with the results of the present study.

The bond strength and nanoleakage values obtained at 24 h and after 6-month water storage for SBU coincide with previous research that reported a similar performance in comparison with CSE [25,30,31,32,33,34,35,36]. It can be attributed not only to the presence of 10-MDP in its composition, but also to the addition of a methacrylate-modified polyalkenoic acid copolymer, known as Vitrebond copolymer (VCP) (1–5%). This copolymer may provide chemical bonding derived from its spontaneous reaction with hydroxyapatite [37].

For both adhesives, nanoleakage percentages remained stable for SBU and CSE. The water-insoluble MDP-Ca salts may have protected the adhesive interface and collagen fibers from degradation. Moreover, this sealing ability even improved for SBU in SE mode probably since collagen fibers were not exposed and, therefore, were not hydrolytically degraded [38]. However, neither SBU nor CSE attained adhesive interfaces hermetically sealed. The resin–dentin interfaces formed with SBU-SE depicted silver ions aligned in a shag-carpet pattern at the transition between the hybrid layer and the adhesive layer similar to the pattern observed for SBU-ER interfaces.

According to the literature, the stability of the bonding interfaces generated with SBU is controversial, as it has been reported to remain stable [31,39,40,41], to be affected by hydrolytic degradation [30,32,40,41], or even to increase after storage when SBU is applied in SE mode [42]. 

XEN was one of the first universal adhesives launched into the market without 10-MDP in its composition, unlike most of this type of adhesives, and is also HEMA-free. These specific characteristics were the reason to select it to evaluate its bonding properties. Overall, XEN attained lower bond strength values and worse sealing capacity than SBU, immediately and after 6 months of water storage, regardless of the adhesive strategy selected, in accordance with Siqueira et al. (2018) [21]. This deficient bonding ability of XEN has been confirmed in a recent 36-month clinical study [7] in non-carious cervical lesions. In this clinical report retention rates of only 55% and 48% were determined when it was applied in SE mode, with and without selective enamel etching, respectively. 

XEN does not contain 10-MDP and its interaction with dentin has been described to occur by an “inverse” functionalized phosphoric acid ester and an acryloylaminoalkyl-sulfonic acid [7,21]. However, chemical bonding to dentin surfaces has not been detected, probably due to the acidic nature of this adhesive, as it is classified as an “intermediate strong” adhesive (pH 1.6) [23], unlike SBU and CSE which are “mild” self-etch adhesives (pH 2.7 [8,23,41] and 2 [25], respectively). The acidic monomers contained in XEN may be able to continue demineralizing the dentin even after polymerization [21], exposing more collagen fibrils and causing greater degradation of the adhesive interface [7].

Moreover, XEN is a HEMA-free adhesive, and other researchers have also reported lower bond strength values for these adhesives in comparison with those that include it [30]. The absence of HEMA in the composition of adhesives may lead to the inclusion of higher concentrations of solvent and water, and difficulties in solvent evaporation, resulting in more blisters in the adhesive layer, as can be evidenced in the SEM image of XEN in accordance with previous studies [9,13]. In contrast, no deleterious effects on bond strength or clinical behavior have been reported in studies that have evaluated other HEMA-free adhesives [43,44].

Regarding the influence of the adhesive strategy on the performance of the universal adhesives tested, no differences were detected for both SBU and XEN when applied in ER or SE mode. This is in line with a recent meta-analysis and a systematic review that concluded that the bond strength to dentin and nanoleakage of multimode adhesives does not depend on the adhesion mode [4,45], nor the risk and intensity of postoperative sensitivity [46]. Conversely, it appears that this trend cannot be completely extrapolated to a clinical situation, as two 5-year clinical trials testing SBU reported lower retention rates in SE mode [47], as well as two recent meta-analyses [16,48]. Moreover, in the case of XEN, this adhesive did not fulfill the ADA criteria for full approval when used in the SE mode after 6- and 36-month recalls [7,24], as mentioned before. 

This trend towards a worse performance for XEN in SE mode is consistent with its lower degree of conversion, despite the absence of HEMA. This could be possibly associated with interference of the acidic monomer conversion, the presence of residual solvents [49], and the phase separation identified that may create excessive differential monomer diffusion into demineralized dentin, especially for this acidic adhesive. Another possible explanation for this could be that XEN includes tertiary butanol as a solvent, which is associated with thinner adhesive layers in comparison with adhesives that use a water-ethanol mixture, such as SBU [19,50]. This could hamper its adequate polymerization due to oxygen inhibition in a significant fraction of its depth [40].

Although in the present study the bonding effectiveness of the universal adhesives SBU and XEN was tested not only immediately but also after 6-month water aging and microtensile data exhibit a good correlation to clinical findings [26], the results cannot be extrapolated to a clinical situation. There are a variety of factors that affect the quality and longevity of the adhesive interfaces in a complex clinical environment, such as the characteristics of the dentin we are bonding to, as caries-affected dentin and sclerotic are common substrates, the operator’s expertise, and the risk of caries of the patient, among others. Nevertheless, in vitro studies are the best option to analyze the mechanism of bonding of the adhesives, of degradation of the interfaces, and to separately evaluate the variables that may affect their bonding effectiveness. The relevance of these findings should be confirmed in randomized clinical trials [51].

## 5. Conclusions

The present study confirms that the bonding performance of universal adhesives to dentin is material dependent. Scotchbond Universal, applied as an ER and a SE adhesive, showed immediate and 6-month aging results comparable to the gold standard self-etch adhesive Clearfil SE Bond, in terms of bond strength and sealing ability. However, Xeno Select exhibited a significantly worse performance, regardless of the evaluation time and adhesion strategy applied. 

## Figures and Tables

**Figure 1 materials-16-03458-f001:**
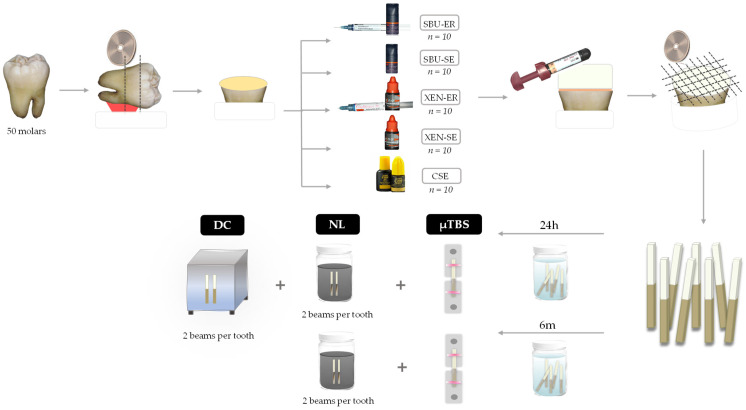
Schematic representation of the experimental design.

**Figure 2 materials-16-03458-f002:**
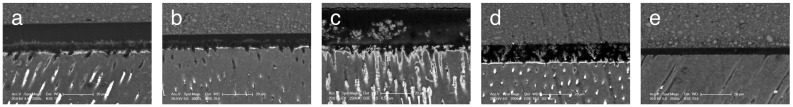
Representative backscattered micrographs of SBU-ER (**a**), SBU-SE (**b**), XEN-ER (**c**), XEN-SE (**d**), and CSE (**e**) nanoleakage at the resin–dentin adhesive interface after 24 h of water storage.

**Figure 3 materials-16-03458-f003:**
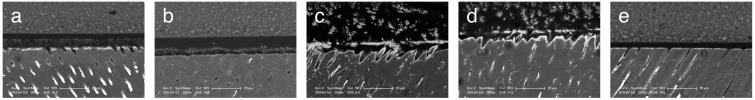
Representative backscattered micrographs of SBU-ER (**a**), SBU-SE (**b**), XEN-ER (**c**), XEN-SE (**d**), and CSE (**e**) nanoleakage at the resin–dentin adhesive interface after 6 months of water storage.

**Table 1 materials-16-03458-t001:** Composition and application techniques, according to the adhesion strategy, of the adhesive systems tested.

Adhesive System	Composition	Etch-and-Rinse Mode	Self-Etch Mode
**Scotchbond Universal (SBU)** (3M Oral Care, St. Paul, MN, USA) Batch number: 523652 pH: 2.7 Universal adhesive	*Etchant (Scotchbond Universal Etchant):* 34% phosphoric acid, water, synthetic amorphous silica, polyethylene glycol, aluminum oxide. *Adhesive:* Bis-GMA(15–25 wt%), HEMA (15–25 wt%), water (10–15 wt%), ethanol (10–15 wt%), silane-treated silica (5–15 wt%), 10-MDP (5–15 wt%), 2-propenoic acid, 2-methyl-reaction products with 1,10-decanediol and phosphorous pentoxide (P_2_O_5_) (1–10 wt%), copolymer of acrylic and itaconic acid (Vitrebond Copolymer) (1–5 wt%), dimethylaminobenzoate (-4) (<2 wt%), (dimethylamino) ethyl methacrylate (<2 wt%), methylethylketone (<0.5 wt%), silane.	Apply the etchant for 15 s. Rinse thoroughly with water for 15 s. Apply the adhesive as in the self-etch mode.	Rub the entire tooth structure for 20 s. Direct a gentle stream of air over the liquid for approximately 5 s, until it no longer moves. Light cure for 10 s.
**Xeno Select (XEN)** (Dentsply Sirona, Konstanz, Germany) Batch number: 1401001210 pH: 1–2 Universal adhesive	*Etchant (Conditioner 36):* 36% phosphoric acid, silica dioxide, detergent, pigment, water. *Adhesive:* Bifunctional acrylates, acidic acrylate, functionalized phosphoric acid ester (ethyl 2-[5-dihydrogen phosphoryl-5,2-dioxapentyl]acrylate), water, tert-butyl alcohol, initiator (camphorquinone), co-initiator (DMABN), stabilizer.	Apply the etchant for 15 s. Rinse thoroughly with water for 15 s. Apply the adhesive as per the self-etch mode.	Rub the entire tooth structure for 20 s. Direct a gentle stream of air over the liquid for approximately 5 s, until it no longer moves. Light cure for 10 s.
**Clearfil SE Bond (CSE)** (Kuraray Noritake Dental Inc., Tokyo, Japan) Batch number: Primer 01109 Bond 01662A pH: 2.0 Two-step self-etch adhesive	*Primer:* 10-MDP, HEMA, camphorquinone, hydrophilic dimethacrylate, N-diethanol N-toluidin-p, water. *Bond:* Bis-GMA, 10-MDP, HEMA, camphorquinone, hydrophilic dimethacrylate, N-diethanol N-toluidin-p, silanated colloidal silica.	Not applicable.	Apply primer to tooth surface and leave in place for 20 s. Blow-dry. Apply bond to the tooth surface and then create a uniform film using a gentle airflow. Light cure for 10 s

Abbreviations: Bis-GMA: bisphenol-A diglycidyl dimethacrylate; HEMA: hydroxyethyl methacrylate; 10-MDP: methacryloyloxydecyl dihydrogen phosphate; DMABN: 4-(dimethylamino)benzonitrile.

**Table 2 materials-16-03458-t002:** Mean (standard deviations) microtensile bond strength (µTBS) values to dentin expressed in MPa and mode of failure (%) for each experimental group (number of teeth tested per group = 10). A: adhesive, CC: cohesive in resin composite, CD: cohesive in dentin, M: mixed.

Adhesive System	24 h		6 Months	
	MPa (SD)	Mode of Failure (%) A/CC/CD/M	MPa (SD)	Mode of Failure (%) A/CC/CD/M
SBU-ER	48.2 (10.4) a1	72/6/22/0	58.3 (4.6) a2	79/11/10/0
SBU-SE	49.9 (20.5) a1	75/10/15/0	50.8 (6.1) a1	83/7/10/0
XEN-ER	22.3 (11.8) b1	94/1/5/0	15.0 (8.9) b2	70/5/25/0
XEN-SE	15.6 (11.9) b1	95/2/3/0	9.3 (6.3) b2	80/4/16/0
CSE	62.7 (8.6) a1	66/10/24/0	59.3 (10.4) a1	77/7/16/0
*p* value	<0.001		<0.001	

* Different lowercase letters within the same column indicate statistically significant differences among experimental groups for each evaluation time (Tukey test). At both evaluation times (24 h and 6 months), “a” means similar bond strength values among SBU-ER, SBU-SE, and CSE, and “b” similar bond strength between XEN-ER and XEN-SE. Groups with letter “a” attained higher µTBS values than groups with “b”. Different numbers in the same row indicate statistically significant differences between 24 h and 6 months of water storage for each adhesive tested (paired *t*-test). XEN-SE and XEN-ER at 6 months (2) exhibited lower µTBS than at 24 h (1). SBU-ER at 6 months (2) showed higher µTBS than at 24 h (1). Similar bond strength values were found for SBU-SE and CSE groups at 24 h (1) and 6 months (1).

**Table 3 materials-16-03458-t003:** Median (interquartile range, IQR) of nanoleakage (NL) expressed in percentage (%) for each experimental group at 24 h and after 6 months of water storage (n = 10). Mean values (standard deviations) expressed in % of degree of conversion (DC) for each experimental group (n = 10).

Adhesive System	Nanoleakage (%)		Degree of Conversion (%)
	24 h Median (IQR)	6 Months Median (IQR)	Mean (SD)
SBU-ER	17.1 (7.0) ab1	18.9 (7.9) ab1	77.2 (16.5) a
SBU-SE	20.7 (9.2) ab2	13.4 (6.9) a1	76.3 (13.7) a
XEN-ER	24.2 (9.9) b1	27.7 (20.6) b1	70.2 (11.2) a
XEN-SE	26.6 (19.4) b1	22.6 (11.9) b1	53.3 (16.5) b
CSE	9.1 (12.5) a1	12.3 (9.8) a1	79.2 (10.8) a
*p* value	0.004	*p* < 0.001	*p* < 0.001

* Different lowercase letters in the same column indicate statistically significant differences among experimental groups for each evaluation time (Mann–Whitney U tests and Bonferroni correction for NL and, Tukey test for DC). Groups with letter “b” showed higher NL values than groups with “a”. Groups with “ab” did not show differences with those with “a” or “b”. At 24 h, “a” means similar NL values among SBU-ER, SBU-SE, and CSE, and “b” similar NL values among XEN-ER, XEN-SE, SBU-ER, and SBU-SE. XEN-ER and XEN-SE (“b”) groups exhibited higher NL values than CSE (“a”). At 6 months, “a” means similar NL values among SBU-ER, SBU-SE, and CSE, and “b” similar NL values between XEN-ER, XEN-SE, and SBU-ER. XEN-ER and XEN-SE (“b”) groups exhibited higher NL values than SBU-SE and CSE (“a”). All adhesive systems showed similar DC (“a”), with the exception of XEN-SE (“b”). Different numbers in the same row indicate statistically significant differences in NL between 24 h and 6 months of water storage for each group (Wilcoxon test). It means that only for SBU-SE, NL after 6 months (1) was lower than after 24 h (2).

## Data Availability

Raw data are available upon request to the corresponding author.

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
