# Peer review of "Long-Term In Vitro Adhesive Properties of Two Universal Adhesives to Dentin"

_materials, 2023, doi:10.3390/ma16093458_

Round 1
Reviewer 1 Report
REVIEWER:
Comments and Suggestions for Authors.
Authors present an in vitro study that evaluated the properties namely the, microtensile bond strength, the nanoleakage and degree of conversion (µTBS, NL, DC) of 2 universal afdhesives, SBU and XEN, applied by 2 modes, ER and SE, respectively;
The aim is relevant for in vitro adhesion evidence to dentin tissue, of those 2 universal. The manuscript is some clear, relevant for the field of materials journal.
About one third of the cited references, belong to the last 5 years.
Moderate editing of English language and style is required;
Reference list and citations, needs to be reviewed, in order to:
* reduce the number substantially the cited references (84 references);
* select and cite the most relevant (in vitro field) and recent references; In this way, authors should try to increment the percentage of cited references of the last 5 years.
* try to avoid, as possible, the self-citations by authors, namely the references number 41, 60, 76, 83.
Title- Recommendation to add the type of study conducted; Please improve the title.
Abstract – Please improve the text. Recommendation to short-describe how the sample was allocated to 2 adhesives, 2 main adhesion modes, and 1 control group;
Main text, materials and methos, results , discussion and conclusions:
Lines 68 and lines 69 - Please try to clear the text, regarding the terms: Usually, Bonding performance is a clinical property, so should not be applied to in vitro studies; Please adequate the sentence, bonding effectiveness to in vitro results;
Line 71- Please explain or re-write the sentence about the composition on HEMA-free and other methacrylic acid-based monomers, of the adhesive Prime&Bond Active. According the manufacture safety data sheet does content is not so clear and is not coherent.
Line 76 to 79 – Suggestion to remove the sentence, once the aim of the study is an in vitro trial, so compactions from different designs and sources should be avoided, in order to avoid information bias.
Line 82- Describe what was the main difference formulation of both experimental adhesives, Only the HEMA content?
Line 86 /Materials and methods/ line 103 and 104/ - Discussion- Experimental groups of the 2 adhesives included the ER and the SE modes. Control group was done by a SE Adhesive (Clearfil SE Bond). As the ER mode was also experimentally performed why does the authors did not used a second control group of the ER mode?
Line 99 /Discussion- 10 samples for each experimental group were performed, for 3 evaluation parameters/properties. Please write how the authors calculated the sample Size for this in vitro trial and discuss if that can be, or not, a limitation of this study according to the main relevant and recent in vitro evidence.
Line 94 and Line 109 – Please explain why were created three incremental layers (2 mm each) once the dentin used was superficial surface (after enamel removed)?
Line 212 to 214 .- Sentence must be removed, Is repeated;
Line 217 to 221 – Please review all the text, and remove repeated sentences.
Discussion:
Describe why the authors choose to test the XENO SELECT and not other universal adhesive?
What criteria did the authors applied to choose those experimental adhesives and the 2 modes, in spite of comparing only the SE mode?
If comparation of dentin properties was to be about HEMA free or not HEMA free (different formulations) adhesives, does another type and design should be implemented.
Line 122 – Table 1/Discussion – Water content (%)of each adhesive. The SBU contains 10-15% of water content; Manufactures instructions of XEN and CSB does not describe the water content of both adhesives. Can the water content of those adhesives formulations interfere to the the µTBS, NL and DC of the adhesives in vitro tested?
Line 410 to 641 .- Restrict and adequate the number of cited and number of references to the main purpose of the properties in vitro studied.
Reviewer 2 Report
Dear Authors,
Please find below some observations and recommendations concerning your article entitled” Long-term adhesive properties to dentin of two universal adhesives applied in etch-and-rinse and self-etch strategies”
Title
Please follow the MDPI authors' guidelines.
In the Abstract section:
Please follow the MDPI authors' guidelines concerning the abstract structure (no more than 200 words should be included, without headings).
- Please add to all commercial products used in your research, details concerning the manufacturer, city, and state.
Eg.
Scotchbond Universal Adhesive (3M ESPE, St Paul, USA)
In the Introduction section:
-Please provide the complete information (manufacturer, city, and state) about the commercial products you referred to
- To make the null hypotheses more understandable, please rewrite them.
Eg. To accomplish the proposed goal, we formulated ”n” null hypotheses:
- first hypothesis
-second hypothesis”
In the Materials and Methods section:
- Please include a diagram of the experimental groups to clarify how the 50 teeth were divided.
- Please take into consideration that table must be included in the text, closest to where it is mentioned- move Table 1 after line 110
-line 111” After storage in distilled water for 24 h at 37ºC ....”- Please move it after Table 1
In the Results section:
-Please take into consideration that tables and figures must be included in the text, closest to where they are mentioned:
1. move Table 2 after line 255” Descriptive data of μTBS for each experimental groups are shown in Table 2.”
2. move Table 3 after line 275” Regardless of the adhesive system and strategy used, all interfaces showed nanoleakage (Table 3).”
3. move Figure 1, 2 after line 285” Representative micrographs of the bonding interfaces for all groups are depicted in Figure 1 and 2.”
- line 255 –” Both universal adhesives showed pre-test failures although the percentage..” move it after table 2
- line 276 -Please make a new paragraph” However, it was significantly...”
- line 277- Please correct (Fig.4): add the figure or remove the text
- line 286 -Please make a new paragraph” Low levels of silver deposits were...”
- Table 2 and 3 please add a row with the ”p values” and provide a legend about the meaning of” a1, b1, a2, b2, ab1....”
In the Discussion section:
- Please add a paragraph about study's limitations
Reviewer 3 Report
The article entitled "Long-term adhesive properties to dentin of two universal adhesives applied in etch-and-rinse and self-etch strategies" is a well-conducted in vitro study. The experimental design is correct, as well as the statistical analysis.
Very interesting is the performance after water aging.
Comparison with other studies has been correctly made in the discussion.
Micrographs are very nice.
The authors could eventually mention clinical consequences of the findings of the study: esthetic issues in direct restorations (PMID: 28983532), lower bonding performances (PMID: 34456050), post-op sensitivity (PMID: 26122377).
Author Response
Thank you very much for your positive comments and for the references suggested that have been included in the new version of the manuscript, except the one regarding the clinical report by Paolone, as it was not considered appropriate.
Round 2
Reviewer 2 Report
Dear authors,
Thank you very much for revising the manuscript according to my comments.
My single recommendation is:
- Table 2 and 3: please mention the full explanation of ” a1, b1, a2, b2, ab1....” in the table footer.
Author Response
Thank you very much for your review. Following your requirement, we have added a more extensive explanation of the lowercase letters and numbers included in table 2 and table 3:
Table 2
* Different lowercase letters within the same column indicate statistically significant differences among experimental groups for each evaluation time (Tukey test). At both evaluation times (24h and 6 months), “a” means similar bond strength values among SBU-ER, SBU-SE and CSE, and “b” similar bond strength between XEN-ER and XEN-SE. Groups with letter “a” attained higher µTBS values than groups with “b”.
Different numbers in the same row indicate statistically significant differences between 24 hours and 6 months of water storage for each adhesive tested (paired t-test). XEN-SE and XEN-ER at 6 months (2), exhibited lower µTBS than at 24 hours (1). SBU-ER at 6 months (2) showed higher µTBS than at 24 hours (1). Similar bond strength values were found for SBU-SE and CSE groups at 24 hours (1) and 6 months (1).
Table 3
* Different lowercase letters in the same column indicate statistically significant differences among experimental groups for each evaluation time (Mann-Whitney U tests and Bonferroni correction for NL and, Tukey test for DC). Groups with letter “b” showed higher NL values than groups with “a”. Groups with “ab” did not show differences with those with “a” or “b”. At 24h, “a” means similar NL values among SBU-ER, SBU-SE and CSE, and “b” similar NL values among XEN-ER, XEN-SE, SBU-ER, SBU-SE. XEN-ER and XEN-SE (“b”) groups exhibited higher NL values than CSE (“a”). At 6 months, “a” means similar NL values among SBU-ER, SBU-SE and CSE, and “b” similar NL values between XEN-ER, XEN-SE and SBU-ER. XEN-ER and XEN-SE (“b”) groups exhibited higher NL values than SBU-SE and CSE (“a”). All adhesive system showed similar DC (“a”); with the exception of XEN-SE (“b”).
Different numbers in the same row indicate statistically significant differences in NL between 24 hours and 6 months of water storage for each group (Wilcoxon test). It means that only for SBU-SE, NL after 6 months (1) was lower than after 24 hours (2).